# Comparative Analysis of Cytokine Expression Profiles in Prostate Cancer Patients

**DOI:** 10.3390/biology14050505

**Published:** 2025-05-06

**Authors:** Karoline Brito Caetano Andrade Coelho, Denise Kusma Wosniaki, Jonatas Luiz Pereira, Murilo Luz, Letusa Albrecht, Jeanine Marie Nardin, Mateus Nobrega Aoki, Leonardo O. Reis, Rodolfo Borges dos Reis, Dalila Lucíola Zanette

**Affiliations:** 1Uro-Oncology Laboratory, Surgery and Anatomy Department, Ribeirao Preto Medical School, University of Sao Paulo, Ribeirão Preto 14090-000, SP, Brazil; karolbcaetano@usp.br (K.B.C.A.C.); rodolforeis@fmrp.usp.br (R.B.d.R.); 2Laboratory for Applied Science and Technology in Health, Carlos Chagas Institute, Oswaldo Cruz Foundation (Fiocruz), Curitiba 81350-010, PR, Brazil; denisewosniaki@gmail.com (D.K.W.); letusa.albrecht@fiocruz.br (L.A.); mateus.aoki@fiocruz.br (M.N.A.); 3Erasto Gaertner Hospital, Curitiba 81520-060, PR, Brazil; jonluizpereira@gmail.com (J.L.P.); muriloaluz@gmail.com (M.L.); 4School of Medicine and Life Sciences, Pontifícia Universidade Católica do Paraná, Curitiba 87013-250, PR, Brazil; jeaninemarie.nardin@gmail.com; 5UroScience, State University of Campinas, Unicamp, Campinas 13083-872, SP, Brazil; 6ImmunOncology, Pontifical Catholic University of Campinas, PUC-Campinas, Campinas 13087-571, SP, Brazil; 7UroGen, National Institute of Science, Technology and Innovation in Genitourinary Cancer (INCT), Campinas 13087-571, SP, Brazil

**Keywords:** prostate cancer, cytokines, interleukins, chemokines, expression, clustering

## Abstract

Specific correlations between IL-4 and MIP-1 beta, IL-4 and IFN-gamma, IL-5 and IL-12p70, and IL-5 and IFN-gamma in PCa patients did not occur in healthy individuals, which might guide future drug targets and candidates with the potential to predict FDA-approved prostate cancer treatment responses by targeting cytokine levels and the oncogenesis pathways.

## 1. Background

Prostate cancer remains a major public health challenge with significant implications for aging male populations globally. The American Cancer Society (ACS) estimates that 299,010 new cases and 35,250 deaths occurred in 2024 alone [1].

Despite advances in the field, ongoing research is critical to improve early detection and risk prediction, develop targeted therapies, reduce health disparities, and enhance survivorship care. Underrepresentation of Brazilian patients in prostate cancer research is a critical and well-documented issue that has important implications for diagnosis, treatment, and health equity [2,3,4].

Chronic inflammation favors the development of all types of cancer, including prostate cancer (PCa) [5]. The tumor microenvironment (TME) is composed of cancer-associated fibroblasts (CAFs), tumor-associated macrophages (TAMs), immune cells, an extracellular matrix (ECM), cancer stem cells, mesenchymal cells, proteins, chemokines, growth factors, and cytokines, which act together to promote tumor survival [6].

Different cytokines secreted by tumor cells contribute to the initiation, progression, and establishment of metastasis in PCa [7] directly by regulating tumor cell growth and metastasis, or indirectly through the modulation of immune and stromal cells and induction of angiogenesis in the TME [8,9]. Cytokines are low-molecular-weight glycoproteins involved in cell communication (autocrine and paracrine) [10], immunity (innate and adaptive), inflammation (pro- and anti-inflammatory), host defense, and tumor immunobiology [11,12,13]. The main classes of cytokines are chemokines, interleukins, interferons, and tumor necrosis factors [14,15].

Chemokines are a large family of cytokines with chemoattractant properties, which are classified into four subclasses according to the first two cysteine amino acid residues: CC, CXC, CX3C, and XC [16]. They regulate different signaling pathways and are therefore associated with numerous biological processes, including epithelial–mesenchymal transition (EMT), growth, angiogenesis, immune evasion, and leukocyte infiltration. They can be produced by the primary tumor and distant metastatic sites [17]. CXCL1, CXCL2, CXCL8, CX3CL1, CCL2, MIP-1α/CCL3, MIP-1β/CCL4, and CCL22 are some of the chemokines that are involved in the development, invasion, and metastasis of PCa. Interleukins (ILs) play a role in signaling and are subdivided into four types and approximately fifty subtypes. IL-1, IL-2, IL-6, IL-7, IL-10, IL-12, IL-15, IL-17, and IL-1β are the main interleukins involved in PCa proliferation [10,13].

Interferons can be classified into three types based on the corresponding receptors, secretory cell, and sequence: (I) IFN-α, β, ε, ω, κ, (II) IFN-γ and (III) IFN-λ [18]. In TME, IFN-γ may be involved in pro-tumorigenic and anti-tumor immunity [19,20]. However, in PCa, the level of IFN-γ is abnormal and appears to play a pro-metastatic role [21,22]. Among the tumor necrosis factors (TNFs), TNF-α is the most commonly reported in cancer patients [13]. In PCa, high levels of TNF-α are associated with increased tumor cell migration and metastatic disease [23,24].

Given the role of cytokines in cancer, those molecules are promising therapeutic targets for interfering with the tumorigenic environment. However, many studies are still needed to advance the understanding of their deregulation and modulation. Although in vitro, in vivo, and pre-clinical studies with cytokines associated with other therapies have shown positive results, developing drugs that target cytokines is still challenging. This study aimed to identify the cytokine expression profile in prostate cancer patients and compare it with data obtained from healthy individuals. Unsupervised machine learning using K-means clustering validated cytokine expression patterns in prostate cancer patients.

## 2. Materials and Methods

### 2.1. Subjects and Samples

This study included seventy-five (n = 75) PCa patients and fourteen (n = 14) healthy men. Plasma samples were obtained from patients from two hospitals, Hospital das Clínicas de Ribeirão Preto (HCRP) in São Paulo/Brazil and Hospital Erasto Gaertner (HEG) in Paraná/Brazil, with proper respective Ethics Committee approvals (CAAE 12905319.1.0000.5440 and CAAE 36931020.9.1001.0098). All subjects signed an informed consent form prior to their inclusion in this study. The clinicopathological data were obtained from medical records. Variables including age at diagnosis, Prostate-Specific Antigen (PSA), TNM staging system, family history of cancer, and ISUP-Grade Group were assessed.

Peripheral blood samples were collected in EDTA tubes before any specific cancer treatment (patients). The blood was centrifuged at 3000 rpm for 10 min to obtain plasma, which was transferred to another tube and centrifuged at 3000 rpm for an additional 10 min. The supernatant from the second plasma centrifugation was stored at −80 °C until it was used for cytokines quantification.

### 2.2. Multiplex ELISA (Luminex)

A customized Multiplex ELISA Kit was used to quantify twelve cytokines: IL-4, IL-5, IL-6, IL-10, IL-1 beta (IL-1β), IL-17A, IL-12p70, MCP-1 (CCL2), MIP-1 alpha (MIP-1α/CCL3), MIP 1-beta (MIP-1β /CCL4), TNF-alpha (TNF-α), and IFN-gamma (IFN-γ) in plasma samples (50uL) from PCa patients and controls. Table 1 illustrates key cytokines’ potential roles, mechanisms, and effects in prostate cancer [13,25,26,27,28,29,30,31,32,33,34,35].

The panels obtained from R&D Systems were Human Discovery Th1/Th2 Mag Luminex Perf Assay Fixed Panel and Human Chemokine Luminex Performance Assay 8-plex Fixed Panel. The fluorescence was detected in the ELISA plate reader based on fluorescent beads according to the manufacturer’s recommendations (Luminex1 MAGPIX1 System; Luminex Corporation, Austin, TX, USA).

### 2.3. Statistical Analysis

Differences in cytokine expression between PCa patients and controls, and analysis stratified by ISUP grade (grades 1–3 vs. 4–5), prostate-specific antigen (PSA; <10 ng/mL vs. ≥10 ng/mL), and TNM stage (T2 M0/N0 vs. M1/N1/T3/T4), were assessed using the Wilcoxon rank-sum test.

The correlation between cytokine expression levels in patients was evaluated using Spearman’s rank correlation (rho). The cohort was clustered using K-means Clustering. The level of significance was established at α = 0.05. All statistical analyses and data visualization were performed using R software (version 4.1.2).

## 3. Results

### 3.1. Clinicopathological Features of PCa Patients

The study included seventy-five men with treatment-free PCa at diagnosis. Fifty percent of the patients were up to 70 years (median) of age with a serum total PSA value of up to 10 ng/dL. According to the TNM Staging System, 57% of the cohort had a palpable tumor confined to the prostate (Tumor Stage T2), three patients had regional lymph node metastasis (N1), and five had distant metastases. The patients’ full clinicopathological data are available in Table 2. All the healthy individuals were male, with an average age of 57 years (range: 34–80 years). We do not have any clinical data available for these individuals.

### 3.2. Comparison of Cytokine Expression Profiles Between Healthy and PCa Men

Twelve cytokines were quantified to identify their differential expression profiles in the plasma of PCa patients and healthy men. Among the measured cytokines, MIP-1 alpha/CCL3, MIP-1 beta/CCL4, IFN-gamma, and interleukins (IL-4, IL-5, IL-6, IL-10, IL-1beta/IL-1F2, IL-17/IL-17A, and IL-12p70) exhibited increased expression in PCa patients. In contrast, TNF-alpha and MCP-1/CCL2 showed elevated expression in the control group. Half of the patients showed MCP-1/CCL2 expression levels of up to 218 pg/mL (median), while the control group had a median of 405 pg/mL. In addition, the control group exhibited higher TNF-alpha expression levels, with a median of 14.9 pg/mL compared to 6.9 pg/mL in the PCa group (Table 3).

When evaluating the interleukins panel, we observed that all of them (IL-4, IL-5, IL-6, IL-10, IL-1beta, IL-17, and IL-12p70) exhibited higher expression in PCa patients compared to healthy individuals (Figure 1). The most pronounced discrepancies in expression profiles between the groups were observed for IL-12p70 (IC95%: 94.6;108.9, *p*-value < 0.0001), IL-4 (IC95%: 33.88; 46.31, *p*-value < 0.0001), IL-5 (IC95%: 4.23; 5.15, *p*-value < 0.0001), and IL-17 (IC95%: 5.94;7.15, *p*-value < 0.0001). The other interleukins exhibited greater variability in expression. The analysis revealed that 50% of healthy individuals showed no detectable IL-4 expression (mean: 1.4; median: 0), whereas IL-4 levels in PCa patients were approximately 30-fold higher (mean: 43; median: 42).

Among the three chemokines quantified (MCP-1, MIP-1 alpha, and MIP-1 beta), MCP-1 was the only one with higher expression levels in the control group, showing high variability. In contrast, MIP-1 alpha (148 pg/mL vs. 4 pg/mL) and MIP-1 beta (240 pg/mL vs. 45 pg/mL) levels were higher in PCa (Figure 2).

Our study found no difference in cytokine expression levels when patients were stratified into two groups based on (a) ISUP Grades (1, 2, and 3 vs. 4 and 5); (b) total PSA levels (PSA < 10 vs. PSA ≥ 10); and (c) TNM stage (T2 M0/N0 vs. M1/N1/T3/T4). In our cohort, patients with higher ISUP grades (4 and 5), indicating a greater risk of aggressive cancer, did not exhibit distinct cytokine expression profiles compared to those with lowerrisk grades. Similarly, patients with higher TNM stage and PSA levels did not show differences in cytokine expression profiles (Appendix A).

### 3.3. Correlation Analysis of Cytokines in PCa Patients

We performed a correlation matrix analysis to investigate potential correlations between the cytokines in the patients’ plasma. The heatmap revealed positive relationships, indicating proportional expression levels. In other words, cytokines in PCa tend to be overexpressed together. Our study found that the correlations between these cytokines were mostly moderate. No negative correlation was found between any pair of cytokines analyzed. The highest associations (rho > 0.7) were observed within the cluster comprising MIP-1 beta, IFN-gamma, IL-12p70, IL-4, and IL-5 (Figure 3).

We demonstrated that the correlations between IL-4 and MIP-1 beta, IL-4 and IFN-gamma, IL-5, and IL-12p70, as well as IL-5 and IFN-gamma, are specific to patients, performing correlational analysis between the cited cytokines for the PCa patients and the control group, as these correlations were nearly absent in healthy individuals (rho ≈ 0).

The high correlation coefficients in patients reflect a strong positive relationship, which was not observed in the healthy controls (Figure 4). Additionally, our sample did not provide evidence of a correlation between total PSA levels and any of the quantified cytokines.

### 3.4. K-Means Clustering: Validation of Cytokine Expression Patterns in PCa Patients

We employed the K-means Clustering algorithm to validate the interdependent relationships in our cohort based on cytokine expression. The variables were standardized using Z-scores and the Euclidean distance was determined. The outputs obtained from Hierarchical Agglomerative Clustering and the Elbow method were applied to determine the optimal number of clusters used as input (Appendix A). The plot indicated that the cohort (PCa patients and controls, n = 89) was grouped into three clusters (two clusters with PCa patients—Clusters 1 and 3—and one cluster with the control group—Cluster 2) (Figure 5) (Appendix A).

The PCa patients were clustered into two groups according to the profiles of twelve cytokines: IL-4, IL-5, IL-6, IL-10, IL-1β, IL-17A, IL-12p70, MCP-1 (CCL2), MIP-1α (CCL3), MIP-1β (CCL4), TNF-α, and IFN-γ. Among the 75 patients, 72 had very similar mean cytokine expression levels, coherent with the clustering pattern. In contrast, the samples from three patients exhibited values which were extremely divergent from the other samples (Cluster 3). These three patients were not included in the main cluster of PCa patients (Cluster 1) due to the atypical expression levels of two cytokines, IL1-beta and MIP1-alpha, compared to the rest of the cohort (PCa patients)**.** Two patients presented atypical expressions (outliers) of MIP1-alpha, and one patient presented an atypical value of IL-1beta (Appendix A). Because of this, Cluster 1 presented an average IL-1 beta level of 11.5 pg/mL, while Cluster 3 had an average of 40 pg/mL.

Regarding MIP-1 alpha, Cluster 3 exhibited an expression approximately six times higher than that of the main PCa patient cluster (Cluster 1). Although three patients formed a separate cluster (Cluster 3), they remain closer to Cluster 1. According to the data available in the medical records, the three patients shared two common characteristics at the time of diagnosis: age >70 (71, 72, and 75 years old) and ISUP Grade 1 and 2 (low–intermediate risk).

Here, the objective was to group observations (individuals) that are internally homogeneous and heterogeneous from each other according to cytokine levels. The clustering analysis confirmed the previous findings, revealing that PCa patients exhibit a distinct expression profile of cytokines IL-4, IL-5, IL-6, IL-10, IL-1β, IL-17A, IL-12p70, MCP-1 (CCL2), MIP-1α (CCL3), MIP-1β (CCL4), TNF-α, and IFN-γ compared to healthy individuals.

Lastly, we applied one-way ANOVA to determine whether cytokine levels differed among the defined clusters. The analysis revealed that all cytokines contributed to forming at least one cluster. IFN-gamma and IL-1beta exhibited the highest F statistics, indicating their most significant contribution to the clustering.

## 4. Discussion

Given the importance of chronic inflammation in cancer, this report compares the levels of a panel of soluble factors, including cytokines, chemokines, and interleukins. PCa patients exhibited higher plasma levels of IL-4, IL-5, IL-6, IL-10, IL-1b, IL-17A, and IL-12p70 than healthy controls.

Chen et al. [36] also reported increased levels of IL-6, IL-10, IFN-γ, and IL-12p70 in Chinese PCa patients compared to prostate hyperplasia. The current study also reports increased levels of IL-6, IL-10, IFN-γ, and IL-12p70 in Brazilian PCa patients, but those were compared to healthy controls, since prostate hyperplasia patients were not included. Also, in agreement with Chen et al., we found decreased TNF-α levels in Brazilian PCa patients compared to healthy controls.

Conversely, we found increased levels of IL-4 and IL-1b in Brazilian PCa patients compared to healthy controls. In contrast, Chen et al. found decreased levels of those molecules in Chinese PCa patients compared to healthy controls [36]. However, IL-1b has also been described as upregulated in North American PCa patients compared to healthy controls in the study of Johnke et al. [37]. The role of IL-1b in cancer is controversial, as it has been reported to induce Th1 and Th17 to strengthen the anti-tumor effect. It has also been shown that both IL-1α and IL-1β may help with tumor angiogenesis and invasiveness as PCa develops, as stated in a review by Ullah, Jiao, and Shen [38].

The main differences between our results and Chen’s study [36], beyond potential ethnic disparities, may be due to the use of serum in their study, compared to plasma in the current study. Rosenberg-Hasson et al. describe that low cytokine levels would only be detected in plasma and not in serum, with a rough correlation of cytokine levels when plasma and serum samples are multiplexed in assays. The discrepancies may be due to the greater inhibitory effects in serum than in plasma, the hemolysis interference, delayed processing, and the use of different anticoagulants. According to the author, using plasma instead of serum might unravel differences at the lower end of the assay’s detection limit that may be concealed by higher background levels in serum samples [39].

In agreement with this, we report the presence of differentially expressed cytokines that were not detected as significant by Chen and colleagues (PCa vs. controls), such as IL-6, IL-10, IL-12p70, and IL-6. Importantly, there are very few reports on cytokine levels in plasma/serum in PCA patients compared to healthy individuals, which limited our discussion compared with Chen’s study [36].

IL-6 has been previously shown to be upregulated in plasma samples from Ugandan PCa patients [40]. IL-6 is overexpressed in localized disease and downregulated in metastatic disease, whilst the opposite was observed for TNFα expression in North Americans in a report by Deichaite et al. [41]. We reported the same inverse pattern for IL-6 and TNF-α expression in our data. Elevated levels of IL-6 have been associated with PCa development and progression, with possible involvement in autocrine and paracrine functions [27]. Despite this, the IL-6 antagonist Siltuximab showed no clinical efficacy in clinical trials in European and North American PCa patients [42].

The increased expression of IL-17A we observed in PCa patients agrees with several previous studies, although most of them analyzed IL-17 expression in the tumor tissue and microenvironment. Conversely, Chen et al. showed that serum IL-17 levels were downregulated in PCa patients [36]. IL-17A stimulates PCa growth and metastasis, even under castration conditions in European PCa patients [43].

This study demonstrated that the anti-inflammatory interleukins IL-4 and IL-10 were upregulated in PCa patients, which agrees with Wise et al.’s [44] research on North Americans and partially with Chen et al. [36], who revealed an upregulation of IL-10 and down-regulation of IL-4 in Chinese PCa patients. As reviewed by Mao et al., IL-4 increases the expression of androgens, activates the JNK pathway, and stimulates tumor progression, while IL-10 inhibits anti-tumor responses and regulates the androgen response [45]. In our report, IL-4 was markedly increased (30-fold) in PCa patients, with mostly undetectable levels of this interleukin in healthy individuals.

IL-1β levels were elevated in PCa patients compared to healthy controls. IL-1β has dual functions in promoting and suppressing tumor progression and cell cycle arrest in PCa. IL-1β can exert anti-tumor effects by preventing metastatic cells from colonizing the metastatic site, thus inhibiting metastasis. On the other hand, IL-1β contributes to tumor angiogenesis and invasiveness in the process of PCa progression [46].

CCL2 levels were significantly decreased in PCa patients compared to healthy controls. CCL2 has been investigated as a diagnostic biomarker in PCa, with a report from Iwamoto et al., which identified increased CCL2 levels in Japanese men with PCa compared to those without prostate cancer [47,48]. Our results were the opposite to the results presented in these reports, but there was a higher variability in CCL2 levels within the PCa group if compared to the other soluble factors analyzed here. MIP-1α/CCL3 and MIP-1β/CCL4 levels were significantly increased in PCa patients compared to healthy individuals. Accordingly, MIP-1β/CCL4 expressions were higher in lesions from patients with PCa and with intraepithelial neoplasia (PIN) lesions than in nonneoplastic prostate patients [49], but to our knowledge, no reports are available regarding MIP-1β/CCL4 plasma levels in PCa patients.

IFN- γ expression levels were increased, while TNF- α levels were decreased in the plasma of PCa patients compared to healthy individuals, in accordance with the observation of Chen et al., which was the only report found about plasma levels of those soluble factors in plasma samples from PCa patients [36]. The scarcity of data for these molecules in plasma samples reinforces the importance of this report.

Correlation analyses showed high associations (rho > 0.7) for IL4 and IFN-γ, IL-4 and CCL4/ MIP-1β, IL-5 and IL-12p70 and IL-5 and IFN-γ. The correlations were not present in the samples of healthy individuals. Reports on prostate cancer are scarce, but Tazaki and colleagues found similar results, with both IL-4 and IFN- γ significantly upregulated in the serum of PCa patients compared to healthy controls. IL-5, IL-12, and IFN-γ were also upregulated in this report, but the authors did not correlate the expression of soluble mediators [50]. IL-4 and IFN-γ play major roles in immune responses, with a mutual antagonistic relationship which has classically been described based on the T helper cell type 2 (Th2) playing a promoting role and Th1 playing an inhibitory role for IL-4. On the other hand, IFN-γ was shown to stabilize Th1 function [51].

The current study focuses on characterizing the levels of various cytokines known to influence prostate cancer pathology in an underrepresented population in the literature [52], exploring the potential interference of important, well-known prognostic factors such as ISUP grade, prostate-specific antigen, and TNM stage in cytokine expression levels, as shown in Appendix A.

Underrepresentation of the Brazilian population poses a critical barrier to equity and precision medicine, reinforcing the importance of localized studies and inclusive global research frameworks. There is a lack of inclusion of Latin American populations in genomic databases, particularly in prostate cancer research, and only a small fraction (<2%) of global genetic studies included Latin American individuals [2]. Also, a study by Pena et al. [3] emphasized the ethnic admixture of the Brazilian population, which includes African, Indigenous, and European ancestry. This complexity is largely unaccounted for in current models of prostate cancer risk and progression, leading to potential misclassification of risk in Brazilian men.

## 5. Study Limitations

Our sample does not provide sufficient evidence to confirm cytokine expression profile differences among individuals with higher PSA levels and/or ISUP grades. However, this limitation may be attributed to the sample size rather than the absence of a true difference. Therefore, studies with larger cohorts are essential to validate these findings. Moreover, we found no evidence of a correlation between PSA levels and cytokine expression in this sample.

## 6. Conclusions

The levels of MIP-1α/CCL3, MIP-1β/CCL4, IFN γ, IL12p70, IL4, IL5, and IL17 were significantly increased, while TNF-α and CCL2 were decreased in PCa patients compared to healthy individuals, indicating a distinct profile in these individuals. Furthermore, our findings provide strong evidence of specific correlations in PCa patients that have not been observed in healthy individuals, particularly between IL-4 and MIP-1 beta, IL-4 and IFN-gamma, IL-5 and IL-12p70, and IL-5 and IFN-gamma.

Beyond the importance of characterizing the underrepresented Brazilian population, current results will guide in vitro and in vivo human prostate cancer drug treatment models, paving the way for exploration of future drug targets and candidates with potential to predict FDA-approved prostate cancer treatment responses by targeting cytokine levels and the oncogenesis pathways.

## Figures and Tables

**Figure 1 biology-14-00505-f001:**
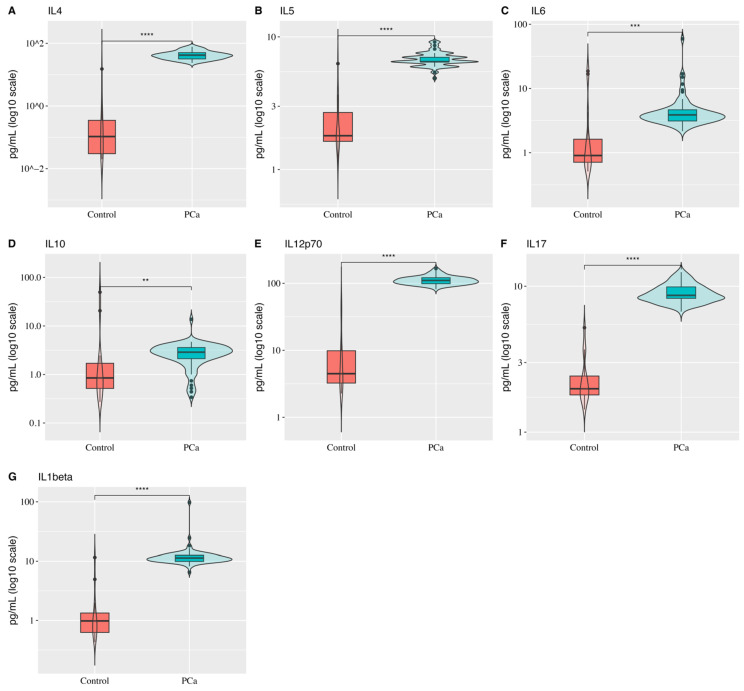
Box-and-violin plots comparing interleukin (IL) expression profiles (pg/mL, log_10_ scale) between healthy controls and prostate cancer (PCa) patients. (**A**) IL-4, (**B**) IL-5, (**C**) IL-6, (**D**) IL-10, (**E**) IL-12p70, (**F**) IL-17, and (**G**) IL-1beta show significantly elevated levels in PCa patients compared to controls. Wilcoxon Test results are indicated as follows: ** *p* < 0.01, *** *p* < 0.001, **** *p* < 0.0001.

**Figure 2 biology-14-00505-f002:**
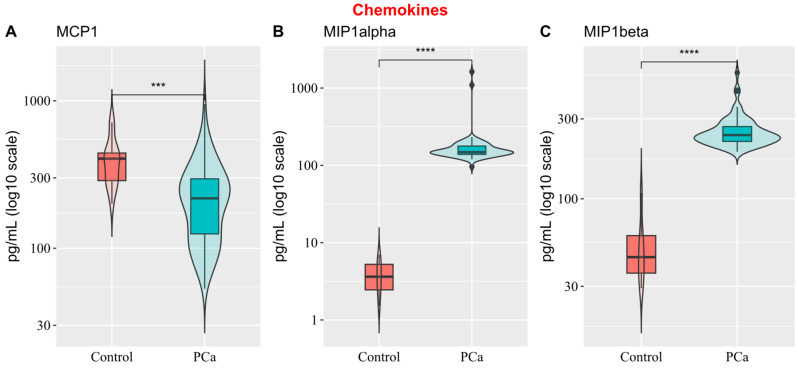
Box-and-violin plots comparing chemokine expression profiles (pg/mL, log_10_ scale) between healthy controls and prostate cancer (PCa) patients. (**A**) MCP1, (**B**) MIP-1 alpha, and (**C**) MIP-1 beta. MIP1alpha and MIP1beta show significantly elevated levels in PCa patients compared to controls, while MCP1 shows decreased levels. Wilcoxon Test results are indicated as follows: *** *p* < 0.001, **** *p* < 0.0001.

**Figure 3 biology-14-00505-f003:**
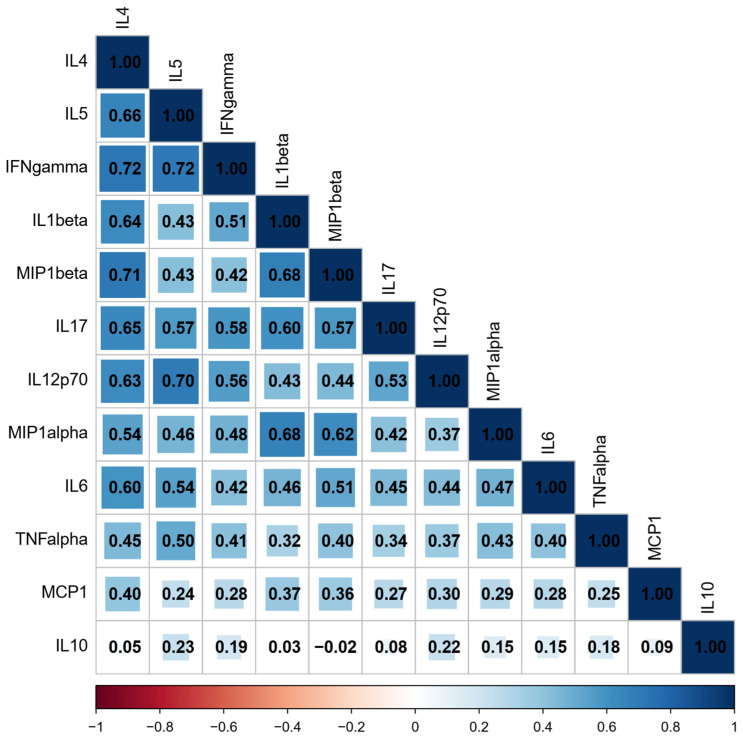
Correlation analysis of cytokines in PCa patients.

**Figure 4 biology-14-00505-f004:**
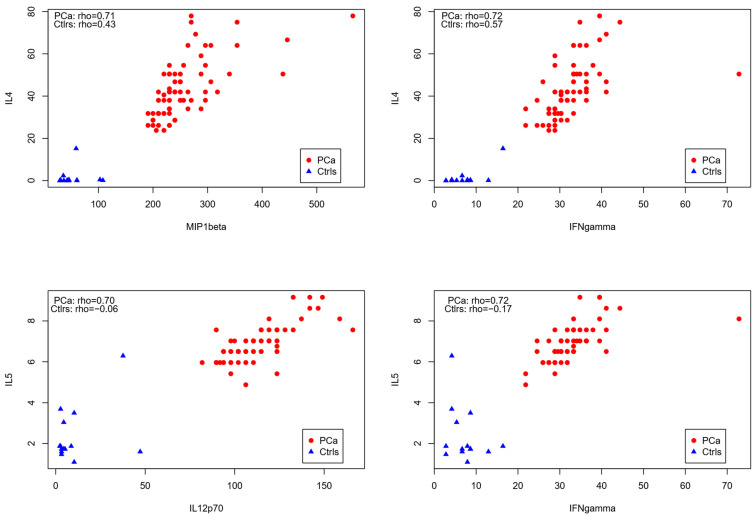
Dispersion correlation analysis of cytokines (pg/dL) in PCa patients and healthy controls.

**Figure 5 biology-14-00505-f005:**
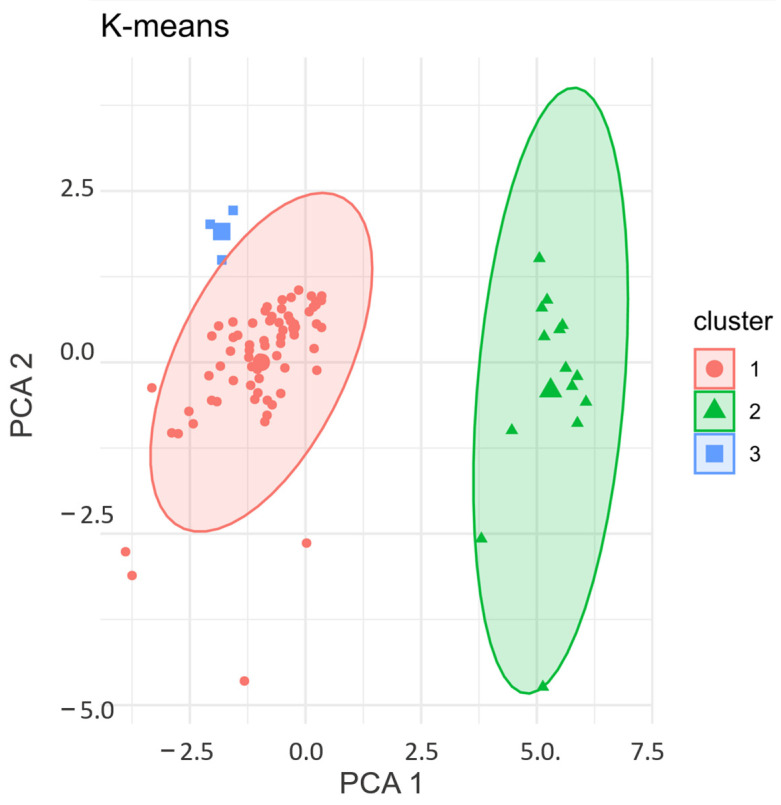
K-Means Clustering: validation of cytokine expression patterns.

**Table 1 biology-14-00505-t001:** Key cytokines’ potential roles, mechanisms, and effects in prostate cancer.

Cytokines	Roles in Prostate Cancer	Mechanisms/Effects	Key Reference
IL-4	Tumor cell survival and resistance	PI3K/AKT pathway; enhances chemo-resistance	[25]
IL-5	Th2-skewed immunity	Reduces cytotoxic responses	[26]
IL-6	PCa progression	AR signaling and STAT3; angiogenesis, castration resistance	[27]
IL-10	Immune-suppressive/dual role	Inhibits T-cell activation; correlates with poor survival	[28]
IL-1β	Invasiveness and treatment resistance	Neuroendocrine differentiation and TME inflammation	[29]
IL-17A	Inflammation/metastasis	Angiogenesis/neutrophil infiltration	[30]
IL-12p70	Anti-tumor immunity	Th1 cells and cytotoxic lymphocytes	[31]
MCP-1/CCL2	Pro-tumoral macrophages	Bone metastasis/recurrence	[32]
MIP-1α/CCL3	Bone metastasis	Osteoclastogenesis	[33]
MIP-1β/CCL4	Immune suppression in TME	Immune cell recruitment to tumor	[13]
TNF-α	Chronic inflammation and resistance	AR-independent tumor growth	[34]
IFN-γ	Antitumor response (context-dependent)	Antigen presentation; PD-L1 upregulation in prolonged exposure	[35]

**Table 2 biology-14-00505-t002:** Patients’ clinicopathological data.

Variables	Patients N = 75 ^1^
Age at diagnosis	70 (65, 72)
BMI	26.2 (24.0, 30.9)
1st PSA	10 (7, 21)
Stage T	
T2	20 (57%)
T3	11 (31%)
T4	4 (11%)
Stage N	
N0	28 (80%)
N1	3 (8.6%)
NX	4 (11%)
Stage M	
M0	31 (86%)
M1	5 (14%)
Family History	
Absent	20 (41%)
Present	29 (59%)
ISUP Grade Group	
1	17 (23%)
2	13 (17%)
3	17 (23%)
4	17 (23%)
5	11 (15%)

^1^ Median (IQR); n (%).

**Table 3 biology-14-00505-t003:** Cytokines’ differential expression profiles in the plasma of PCa patients and healthy men (control).

	Control	PCa
	N = 14 ^1^	N = 75 ^1^
MCP1	405 (281, 442)	218 (125, 296)
MIP1alpha	4 (2, 5)	148 (139, 178)
MIP1beta	45 (36, 61)	240 (220, 270)
IFNgamma	7 (4, 9)	32 (29, 35)
IL10	0.86 (0.50, 1.84)	2.88 (2.12, 3.64)
IL12p70	4 (3, 10)	110 (98, 124)
IL4	0 (0, 0)	42 (32, 50)
IL5	1.80 (1.60, 3.04)	6.50 (6.50, 7.02)
IL6	0.9 (0.7, 1.7)	3.9 (3.1, 4.7)
TNFalpha	14.9 (11.6, 18.1)	6.9 (6.0, 7.9)
IL1beta	1.0 (0.6, 1.4)	11.3 (9.9, 12.7)
IL17	1.99 (1.80, 2.46)	8.66 (8.26, 9.90)

^1^ Median of cytokine differential expression, pg/dL (Q1, Q3).

## Data Availability

The original contributions presented in this study are included in the article/Appendix A. Further inquiries can be directed to the corresponding authors.

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
