# Peer review of "Comparative Analysis of Cytokine Expression Profiles in Prostate Cancer Patients"

_biology, 2025, doi:10.3390/biology14050505_

Round 1

Reviewer 1 Report

Comments and Suggestions for Authors

The manuscript titled ‘Comparative Analysis of Cytokine Expression Profiles in Prostate Cancer Patients' is comparing the plasma samples of PCa patients with healthy individuals in terms of 12 different cytokines using Multiplex ELISA. The results are represented in a way that’s easy to visualize and understand easily for a person who is not related to the field. However, this work has some major flaws that needs to be addressed in order for a complete understanding of prostate pathology.  

Major Concerns:

The authors aim to compare the levels of various cytokines known to influence prostate cancer pathology. As the authors pointed out, multiple studies have already examined cytokine levels in prostate cancer (PCa) across different human samples. However, Coelho et al.'s article would greatly benefit from incorporating novel insights into prostate cancer research. Specifically, exploring the impact of FDA-approved prostate cancer treatments, or potential drug candidates targeting the cancerogenesis pathway, would provide crucial information on how these therapies can reduce both cancer severity and cytokine levels. For this purpose, the authors should also consider utilizing in vitro cell culture models of human prostate cancer cells for drug treatment models. Currently, this manuscript merely compares cytokine levels between the PCa group and the control, reflecting findings already established in previous studies. To enhance the manuscript’s relevance and significantly amplify its impact on the scientific community, the authors should include a comparison group with drug treatments, offering a more comprehensive understanding of how these therapies may modulate cytokine profiles and improve patient outcomes.

Additionally, the authors have not provided a justification for the observed increase or decrease in the levels of certain cytokines. The study does not explain how these changes in cytokine expression in prostate cancer contribute to the differences observed between the control and experimental groups

Minor Concerns:

Please consider including more published studies from the literature to provide a comprehensive background on the existing research in this field. Additionally, presenting data on the prevalence of prostate cancer in men from recent years would be a strong way to emphasize the importance of studying this cancer in the introduction.

It would be helpful to indicate what the roles of these different cytokines are and how their expression levels associated with the progression of cancer

Line 239: Missing reference for this finding from earlier

Author Response

Reviewer 1

The manuscript titled ‘Comparative Analysis of Cytokine Expression Profiles in Prostate Cancer Patients' is comparing the plasma samples of PCa patients with healthy individuals in terms of 12 different cytokines using Multiplex ELISA. The results are represented in a way that’s easy to visualize and understand easily for a person who is not related to the field. However, this work has some major flaws that needs to be addressed in order for a complete understanding of prostate pathology.  

A: Thank you for your constructive comments.

Major Concerns:

The authors aim to compare the levels of various cytokines known to influence prostate cancer pathology. As the authors pointed out, multiple studies have already examined cytokine levels in prostate cancer (PCa) across different human samples. However, Coelho et al.'s article would greatly benefit from incorporating novel insights into prostate cancer research. Specifically, exploring the impact of FDA-approved prostate cancer treatments, or potential drug candidates targeting the cancerogenesis pathway, would provide crucial information on how these therapies can reduce both cancer severity and cytokine levels. For this purpose, the authors should also consider utilizing in vitro cell culture models of human prostate cancer cells for drug treatment models. Currently, this manuscript merely compares cytokine levels between the PCa group and the control, reflecting findings already established in previous studies. To enhance the manuscript’s relevance and significantly amplify its impact on the scientific community, the authors should include a comparison group with drug treatments, offering a more comprehensive understanding of how these therapies may modulate cytokine profiles and improve patient outcomes.

A: Thank you for the thoughtful comments. We have highlighted in the discussion section that “the current study focuses in characterizing the levels of various cytokines known to influence prostate cancer pathology in an underrepresented population in the literature [x], exploring the potential interference of important well known prognostic factors such as ISUP grade, prostate-specific antigen, and TNM stage in cytokine expression levels, supplementary Table 1. By characterizing the Brazilian population regarding clinically relevant cytokine levels our study will guide our next in vitro and in vivo human prostate cancer drug treatment models, paving the way for exploration of future drug targets and candidates with potential to predict FDA-approved prostate cancer treatments response by targeting cytokine levels and the oncogenesis pathways”

Additionally, the authors have not provided a justification for the observed increase or decrease in the levels of certain cytokines. The study does not explain how these changes in cytokine expression in prostate cancer contribute to the differences observed between the control and experimental groups

A: We have expanded the discussion section to “explore how changes in cytokine expression in prostate cancer might play a role in the near future, contributing to guide our next in vitro human prostate cancer cell cultures for drug treatment models, paving the way for future exploration of potential drug candidates, or the impact of FDA-approved prostate cancer treatments, providing crucial information on how these therapies can reduce both cancer severity and cytokine levels by targeting the oncogenesis pathways.” The potential role of these different cytokine expression levels in the PCa progression has also been explored.

Minor Concerns:

Please consider including more published studies from the literature to provide a comprehensive background on the existing research in this field. Additionally, presenting data on the prevalence of prostate cancer in men from recent years would be a strong way to emphasize the importance of studying this cancer in the introduction.

A: We have expanded a comprehensive background on the existing research in this field and on the importance of characterizing the underrepresented Brazilian population in the literature.

It would be helpful to indicate what the roles of these different cytokines are and how their expression levels associated with the progression of cancer

A: We have added Table 1, which illustrates key cytokines' potential roles, mechanisms, and effects in prostate cancer.

Line 239: Missing reference for this finding from earlier

A: Reference added.

Reviewer 2 Report

Comments and Suggestions for Authors

The study “Comparative analysis of cytokine profiles..” by Coelho et al, is focused on data outcomes and correlations from the plasma cytokine profiling of prostate cancer patients with healthy individuals.

Though the author team has put in great efforts to put this study together and analyze the data outcomes-there are several shortcomings.

Major Comments:

It is not clear what the inference of this study is! One major flaw is that study variables have not been considered during data analysis. For example, outcomes from plasma analysis of all the PCa patients have been grouped as one cohort and compared with healthy individuals; however, the PCa patients as per their clinical pathological grading are at different stages of PCa progression. As such, there is a strong possibility that their plasma cytokine profiles would vary. The authors might want to reanalyze their data by grouping the outcomes of the patients whose pathological grades are similar and then assess stage specific profiles. As of now, profiling all patient data together generates no concrete finding; there are more unanswered questions than answers-the study observations will not be reproducible and add anything substantial to future literature (in its current presentation). If PSA levels were not indicative of any difference in cytokine profile, did the authors consider PSA doubling time? The age range of healthy individuals seems to be mostly on the younger side compared to the patient population, that could be another reason for skewed analysis.

Author Response

Reviewer 2

The study “Comparative analysis of cytokine profiles..” by Coelho et al, is focused on data outcomes and correlations from the plasma cytokine profiling of prostate cancer patients with healthy individuals. Though the author team has put in great efforts to put this study together and analyze the data outcomes-there are several shortcomings.

A: Thank you for your constructive comments.

Major Comments:

It is not clear what the inference of this study is! One major flaw is that study variables have not been considered during data analysis. For example, outcomes from plasma analysis of all the PCa patients have been grouped as one cohort and compared with healthy individuals; however, the PCa patients as per their clinical pathological grading are at different stages of PCa progression. As such, there is a strong possibility that their plasma cytokine profiles would vary. The authors might want to reanalyze their data by grouping the outcomes of the patients whose pathological grades are similar and then assess stage specific profiles. As of now, profiling all patient data together generates no concrete finding; there are more unanswered questions than answers-the study observations will not be reproducible and add anything substantial to future literature (in its current presentation). If PSA levels were not indicative of any difference in cytokine profile, did the authors consider PSA doubling time? The age range of healthy individuals seems to be mostly on the younger side compared to the patient population, that could be another reason for skewed analysis.

A: Thank you for the thoughtful comments.

  • The authors thank for this comment and apologize if it was not clear in the manuscript. In fact, on line 221 of the revised article we report that the analyses were carried out, but we do not report the results as figures or tables. Here follows the sentence on line 221 of the manuscript in which we describe these results: “Our study found no difference in cytokine expression levels when patients were stratified into two groups based on: a) ISUP Grades (1, 2, and 3 vs. 4 and 5) and b) total PSA levels (PSA < 10 vs. PSA ≥ 10). In our cohort, patients with higher ISUP grades (4 e 5), indicating a greater risk of aggressive cancer, did not exhibit distinct cytokine expression profiles compared to those with lower-risk grades. Similarly, patients with higher PSA levels did not show differences in cytokine expression profiles.”
  • The cytokine expression levels were also analyzed in the groups of patients stratified by ISUP grade (grades 1–3 vs. 4–5), prostate-specific antigen (PSA; <10 ng/mL vs. ≥10 ng/mL), and TNM stage (T2 M0/N0 vs. M1/N1/T3/T4) using the Wilcoxon rank-sum test. PSA and ISUP grade data were available for all patients, whereas analyses based on TNM staging were performed in a subset of 46 patients (Supplementary_Table_2). We have included this analysis in the manuscript, highlighted in yellow.

Reviewer 3 Report

Comments and Suggestions for Authors

Please go through the attached word file for suggestion.

Author Response

Reviewer 3

Abstract should be about 200 words maximum. Should be single paragraph without headings.

A: Thank you. We have edited the abstract accordingly.

Reference numbers should be placed in square brackets [] and placed before punctuation. In reference abbreviated journal name and volume should be in italics.

A: We have edited the references accordingly.

Line 53-54: Follow a uniform reference style.

A: Corrected.

Line 134: Table 2, mention what are the values in parentheses and the unit of values.

A: Table 2 legends inform what the values in parentheses are and the units of measurement.

Line 145: Explain the plot in Figure 1 to make it self-explanatory and do the same for others.

A: Figures have been made self-explanatory.

Line 163: It has been mentioned no negative correlation between citokynes, but in Figure 3, IL10 and MIP 1 beta have negative correlation. Is there any reason to not considering this pair?

A: We thank the reviewer for the observation. As shown in the heatmap (Fig. 3), the Spearman correlation coefficient between IL-10 and MIP1beta is negative, with a rho value of –0.02. This coefficient was calculated to assess the presence of a monotonic relationship between the two variables. However, given that the rho value is close to zero, it indicates no association.

This interpretation is consistent with Zou, Tuncali and Silverman (DOI: 10.1148/radiol.2273011499), who note that correlation coefficients near zero reflect the absence of an association (see Table 1).

Therefore, this correlation was not considered further in the analysis, as the observed value does not support a statistically or biologically relevant relationship between IL-10 and MIP1beta.

Line 236: Please clarify “authors’ use of serum”. Consider changing the sentence.

A: The sentence was edited.

The main question in the research?

Explores the cytokines that are expressed in prostate cancer as compared to normal healthy individuals. This will be helpful to find biomarkers that specially increase or decrease in prostate cancer patients.

A: Thank you for the constructive comments.

We have also expanded to “explore how changes in cytokine expression in prostate cancer might play a role in the near future, contributing to guide our next in vitro human prostate cancer cell cultures for drug treatment models, paving the way for future exploration of potential drug candidates, or the impact of FDA-approved prostate cancer treatments, providing crucial information on how these therapies can reduce both cancer severity and cytokine levels by targeting the oncogenesis pathways.” The potential role of these different cytokines’ expression levels in the PCa progression has also been explored.

Round 2

Reviewer 1 Report

Comments and Suggestions for Authors

The authors have addressed the comments and incorporated the information as needed within the text. Although the major concern was about performing new sets of experiments in their experimental group, the authors provided justification for what their study wanted to highlight and what the future study would follow. The manuscript can be accepted in this format

Reviewer 2 Report

Comments and Suggestions for Authors

Comments have been addressed.